# House Dust Mite Induces Bone Marrow IL-33-Responsive ILC2s and T_H_ Cells

**DOI:** 10.3390/ijms21113751

**Published:** 2020-05-26

**Authors:** Emma Boberg, Kristina Johansson, Carina Malmhäll, Julie Weidner, Madeleine Rådinger

**Affiliations:** Krefting Research Centre, Internal Medicine and Clinical Nutrition, Institute of Medicine, Sahlgrenska Academy, University of Gothenburg, SE-40530 Gothenburg, Sweden; emma.boberg@gu.se (E.B.); kristina.johansson@ucsf.edu (K.J.); carina.malmhall@gu.se (C.M.); julie.weidner@gu.se (J.W.)

**Keywords:** IL-33, ST2, asthma, allergy, IL-5, innate lymphoid cells, T helper cells, type 2 inflammation, eosinophilia, adaptive immunity

## Abstract

Type 2 innate lymphoid cells (ILC2s) and their adaptive counterpart type 2 T helper (T_H_2) cells respond to interleukin-33 (IL-33) by producing IL-5, which is a crucial cytokine for eosinophil development in the bone marrow. The aim of this study was to determine if bone marrow ILC2s, T_H_ cells, and eosinophils are locally regulated by IL-33 in terms of number and activation upon exposure to the common aeroallergen house dust mite (HDM). Mice that were sensitized and challenged with HDM by intranasal exposures induced eosinophil development in the bone marrow with an initial increase of IL5Rα^+^ eosinophil progenitors, following elevated numbers of mature eosinophils and the induction of airway eosinophilia. Bone marrow ILC2s, T_H_2, and eosinophils all responded to HDM challenge by increased IL-33 receptor (ST2) expression. However, only ILC2s, but not T_H_ cells, revealed increased ST2 expression at the onset of eosinophil development, which significantly correlated with the number of eosinophil progenitors. In summary, our findings suggest that airway allergen challenges with HDM activates IL-33-responsive ILC2s, T_H_ cells, and eosinophils locally in the bone marrow. Targeting the IL-33/ST2 axis in allergic diseases including asthma may be beneficial by decreasing eosinophil production in the bone marrow.

## 1. Introduction

Type 2 innate lymphoid cells (ILC2s) are effector cells typically located at mucosal surfaces where they respond rapidly to environmental insults such as allergens in an antigen-independent manner [1,2,3]. Studies of inflammatory diseases including asthma, atopic dermatitis, chronic rhinosinusitis, and viral or helminth infections have revealed that ILC2s play important roles in various pathological contexts [4,5,6,7,8,9,10,11,12,13]. ILC2s develop in the bone marrow from common lymphoid progenitors and progress into ILC precursors that generate ILC1, ILC2, and ILC3 subsets [14]. Airway allergen challenge triggers the release of epithelial-derived interleukin (IL)-33, which activates cells that express the IL-33 receptor (ST2) such as ILC2s and CD4^+^ T helper (T_H_) cells, including type 2 T helper cells (T_H_2) and regulatory T cells [15,16,17,18]. Although ST2 is strongly associated with the function of T_H_2 cells [19], it has been shown that T_H_1 cells can also transiently express ST2 upon differentiation during viral infection [20]. In contrast to ILCs, T cells are a part of the adaptive immune system and are characterized by their ability to recognize antigens by their specific T cell antigen receptor. Cytokine production by T cells is one of the key characteristics of IL-33 [21]. Both T_H_2 cells and ILC2s produce type 2 cytokines such as IL-4, IL-5, and IL-13 upon activation at sites of inflammation [14]. We have previously shown in an IL-33-induced mouse model of type 2 airway inflammation that bone marrow ILC2s locally produce IL-5, which is essential in eosinophil biology [22] controlling crucial features such as cell proliferation, migration, and the differentiation of eosinophils [23,24,25]. Furthermore, eosinophilia is a key feature in allergic diseases including asthma [26]. Eosinophils develop from CD34^+^ progenitor cells in the bone marrow in the presence of IL-3, granulocyte-macrophage colony-stimulating factor (GM-CSF), and IL-5 [27,28], and eosinophils expressing the chemokine receptor CCR3 can be recruited to the airways during inflammatory conditions [29]. Thus, it is important to study disease mechanisms in the bone marrow during allergic conditions. Several currently available biological therapies for eosinophilic diseases including asthma target IL-5 or its receptor [30]. However, this treatment is inefficient in some individuals with eosinophilic asthma [31,32]. We have previously shown that a higher number of mature eosinophils express ST2 in a direct IL-33-induced asthma model compared to control mice [22] and we recently observed a higher number of ST2^+^ eosinophils and IL-33-responsive ILC2s in the bone marrow in an IL-33- dependent acute protease allergen model [33]. Additionally, it has been shown that the type 2 cytokine response and, consequently, eosinophilic inflammation was decreased in IL-33 receptor deficient mice when exposed to allergen challenge [34,35]. Furthermore, an increased number of ST2^+^ eosinophils in both sputum and blood was observed in asthmatic individuals after allergen challenge [36]. Currently, there is a lack of studies examining ILC2s, T_H_ cells, and eosinophils simultaneously in the bone marrow during eosinophilic inflammation after airway allergen challenge. In addition, IL-33-dependent mechanisms in the bone marrow post-allergen challenges are understudied. Therefore, the aim of this study was to determine if bone marrow ILC2s, T_H_ cells, and mature eosinophils were responsive to IL-33 in mice exposed to the common and clinically relevant aeroallergen house dust mite (HDM). Our data suggest a critical role for IL-33-responsive ILC2s at the onset of HDM-induced eosinophil development in the bone marrow.

## 2. Results

### 2.1. Airway Exposure to House Dust Mite Induced Eosinophil Development in the Bone Marrow

To study eosinophil development in the bone marrow during allergic airway inflammation, mice were only sensitized (HDM/phosphate buffer saline (PBS)) or both sensitized and challenged (HDM/HDM) by repeated intranasal doses of the common aeroallergen HDM (Figure 1A). Elevated levels of bronchoalveolar lavage (BAL) eosinophils were observed exclusively in HDM/HDM mice compared to vehicle treated control mice (PBS/PBS) and HDM/PBS mice (Figure 1B) when assessed by differential cell count. A corresponding increase of the eosinophil-recruiting chemokine CCL24/Eotaxin-2 was detected in BAL from HDM/HDM mice (Figure 1C), suggesting recruitment of eosinophils from the bone marrow to the airways in response to allergen challenge. Eosinophils develop from eosinophil progenitors in the bone marrow and are recruited to sites of inflammation. To distinguish the developmental stage of eosinophils in the bone marrow, flow cytometric analysis was used. The total cell number in the bone marrow was similar among the three groups (Figure 1D), and the number of eosinophil progenitors (SSC^lo^CD45^+^CD34^+^IL5Rα^+^) increased after allergen sensitization (HDM/PBS) compared to control mice (PBS/PBS) (Figure 1E,F). When subsequently challenged with HDM (HDM/HDM), the number of eosinophil progenitors decreased to baseline levels (Figure 1E,F) and a significantly increased number of mature eosinophils was observed (Figure 1G,H). These data suggest that airway exposure to HDM promotes eosinophil hematopoiesis in the bone marrow, where an early induction of eosinophil progenitors is followed by eosinophil maturation. IL-5 is known to be crucial for eosinophil maturation and survival, and more recently, IL-33 has been implicated in this process by upregulating IL-5 receptor alpha (IL5Rα) expression [37]. We next analyzed the expression of the IL-33 receptor ST2 on eosinophils in the bone marrow from HDM/HDM mice and found an increased number of ST2^+^ eosinophils compared to PBS/PBS control mice (Figure 1I). This implicates IL-33 in the regulation of allergen-induced bone marrow eosinophilia and opens the possibility that IL-33 activates eosinophils and other immune cells in the bone marrow that express ST2, which may contribute to eosinophilic airway inflammation. 

### 2.2. IL-33-Responsive Bone Marrow ILC2s, but Not T_H_ cells, Correlated with the Onset of Allergen-Induced Eosinophilia

We recently demonstrated that bone marrow ILC2s support IL-33-driven eosinophil development in the bone marrow after direct intranasal challenges with IL-33 [22] and that ILC2s respond to IL-33 during acute IL-5-dependent eosinophilic inflammation trigged by intranasal exposures to the allergen protease papain [33]. It is unknown whether bone marrow ILC2s respond to IL-33 released upon airway challenges of the common allergen HDM. Increased ST2 expression on ILC2s was observed in sensitized HDM/PBS mice compared to control mice or HDM/HDM mice (Figure 2A). We next examined whether bone marrow ILC2s may influence eosinophil development in the bone marrow. The ST2 expression on ILC2s showed a positive correlation with the number of eosinophil progenitors (Figure 2B). In contrast, a negative correlation was observed between ST2 expression on ILC2s and mature eosinophils (Figure 2C). These data suggest a critical role for IL-33-responsive ILC2s at the onset of allergen-induced eosinophil development in the bone marrow. Furthermore, the number of bone marrow ILC2 was decreased in both HDM/PBS and HDM/HDM mice compared to PBS/PBS control mice (Figure 2D,E), indicating that ILC2 might leave the bone marrow and migrate to the airways upon allergen challenge, which has been previously reported [38].

Although IL-33-responsive T_H_ cells have been studied in many diseases including asthma, currently there is a lack of studies examining T_H_ cells that respond to IL-33 in the bone marrow. We next assessed the number of T_H_ cells in the bone marrow, which was found to be similar in all experimental groups (Figure 3A,B). In contrast to ILC2s, not all T_H_ cells in the bone marrow express the ST2 receptor. However, airway allergen challenges resulted in a significantly increased number of ST2^+^ T_H_ cells in both HDM/PBS and HDM/HDM mice compared to PBS/PBS control mice (Figure 3C). Moreover, the intensity of ST2 expression was significantly increased only in HDM/HDM compared to PBS/PBS mice (Figure 3D). In contrast to ILC2s, the ST2 expression on T_H_ cells did not correlate with the number of eosinophil progenitors or mature eosinophils (Figure 3E,F). Bone marrow ILC2s and T_H_ cells both respond to IL-33 by modulating ST2 expression after airway allergen challenge. Our data suggest that ILC2s are the predominant IL-33-responsive cell type active at the onset of allergen-induced eosinophilia, as indicated by the correlation between ST2 expression on ILC2s and the induction of eosinophil progenitors.

### 2.3. Synergistic Effect of HDM and IL-33 on IL-5 Production by Bone Marrow ILC2s Ex Vivo

IL-5 is critical in the regulation of eosinophil production, maturation, and survival in the bone marrow. We investigated whether HDM and IL-33 stimulations ex vivo induced the production of IL-5 in ILC2s and T_H_ cells. Additionally, we examined if cells from allergen-challenged mice responded differently compared to cells from control mice. IL-33 stimulation induced IL-5 production by ILC2s, as previously demonstrated (Figure 4A) [22,33]. Notably, the combination of HDM+IL-33-induced IL-5 production by ILC2s to a significantly higher level than IL-33 alone (Figure 4A). The intensity of IL-5, measured by mean fluorescence intensity (MFI), was increased in ILC2s in bone marrow cultures stimulated with HDM+IL-33 compared to IL-33 only (Figure 4B), whereas HDM alone did not induce IL-5 production by ILC2s (Figure 4A). Moreover, none of the stimuli induced IL-5 production by T_H_ cells (Figure 4C). The bone marrow cells from PBS/PBS and HDM/HDM responded equally to each condition (Figure 4A–C). Collectively, these data indicate that a potential synergistic effect of HDM and IL-33 on IL-5 production by ILC2s may exist. Future studies are needed in order to explore the biological impact that this synergy might have on eosinophilic inflammation. In line with our previous data, we show that bone marrow ILC2s are an important source of IL-5 and that these cells are not mirroring the function of their adaptive counterpart T_H_2, which did not produce IL-5 under the examined conditions.

## 3. Discussion

There is substantial evidence that IL-33, a member of the IL-1 family of cytokines, has effects on both the innate and adaptive immune responses [39,40]. A large number of immune cells express the IL-33 receptor ST2, including macrophages, mast cells, basophils, eosinophils, neutrophils, T_H_2 cells, regulatory T cells, and ILC2s [15,16,17,18]. Several studies have investigated inflammatory mechanisms in terms of IL-33 regulation during allergic inflammation in the airways. Studies of allergen-induced airway inflammation using IL-33/ST2 knockout mice or neutralizing antibodies against IL-33/ST2 reported decreased type 2 inflammation and lower eosinophilic inflammation as a consequence [6,41]. In the current study, we demonstrate an increased number of ST2^+^ mature eosinophils in the bone marrow after airway exposures to the common allergen HDM, which affects millions of allergic asthmatic individuals worldwide [42]. These results concur with our previous results showing increased numbers of ST2^+^ mature eosinophils after an acute protease allergen model using papain [33]. These data further strengthen the hypothesis that mature eosinophils are IL-33-responsive locally in the bone marrow post-intranasal allergen challenges. Interestingly, a clinically relevant human study has reported that the number of ST2^+^ eosinophils in both sputum and blood was increased in allergic asthmatics that underwent allergen challenge, arguing for a significant role for IL-33 in eosinophil biology [43].

IL-33-responsive ILC2s have been extensively studied during the past years and are discussed as novel therapeutic targets in IL-33-mediated diseases, including human airway diseases [44,45,46]. We identified increased ST2 expression on bone marrow ILC2s in mice that were only sensitized with HDM compared to both PBS controls and mice that were sensitized and challenged with HDM. This increased ST2 expression in HDM-sensitized mice coincided with early eosinophil induction. Moreover, ST2 expression on ILC2s from HDM-sensitized and HDM-sensitized and challenged mice strongly correlated with the numbers of eosinophil progenitors (Figure 2B), whereas a negative correlation of ST2 expression on ILC2s and the number of mature eosinophils was observed. Collectively, these data suggest a critical role for IL-33-responsive ILC2s specifically at the onset of allergen-induced eosinophil development in the bone marrow as a result of airway exposures to HDM. Moreover, we show that this role may be unique for ILC2s compared to T_H_ cells in the bone marrow (Figure 3E,F), since ST2 expression on T_H_ cells did not correlate with the numbers of eosinophil progenitors nor mature eosinophils. Nevertheless, a significant increase in ST2-expressing T_H_ cells was observed in the bone marrow from both HDM-sensitized and challenged mice compared to control mice. These findings suggest that T_H_ cells can contribute to the ongoing inflammation via the IL-33/ST2 axis locally in the bone marrow. Indeed, IL-33 responsive T_H_ cells have been reported in several studies of allergic diseases including asthma [21]. However, a limitation of the present study is that IL-33-responsive bone marrow T_H_2 cells specifically were not analyzed. Therefore, prospective studies are needed to determine the exclusive role of bone marrow T_H_2 cells compared to their innate counterpart ILC2s regarding the effects from IL-33 during type 2 inflammation under allergic conditions.

The number of bone marrow ILC2s was decreased in mice only sensitized or sensitized and challenged with HDM possibly due to ILC2s exiting the bone marrow and migrating toward the airways. This migration has been previously reported in a study demonstrating that IL-33 promoted egress of ST2^+^ ILC2 progenitor cells from the bone marrow [38]. Furthermore, a decreased number of ST2^+^ ILC2 progenitor cells in the bone marrow after intranasal allergen challenges of the fungal *Alternaria* was reported [38]. However, future studies are needed in order to clarify this mechanism in HDM-induced asthma. These results were in contrast to our previous study of acute allergen induction by papain where a higher number of bone marrow ILC2s was observed [33]. Studies investigating the kinetics of mechanism(s) of action are needed in order to elucidate the complete role of bone marrow ILC2s in terms of their migration capacity post-airway challenges of specific allergens and, more importantly, their clinical relevance.

The bone marrow is a key tissue for the production of leucocytes. The bone marrow support cell differentiation, mobilization, and local differentiation of eosinophils progenitors, and this process supports airway eosinophilia in asthmatic patients [47,48]. The cytokine IL-5 promotes eosinophilic inflammation, and we have previously shown that ILC2s and not T_H_ cells produce IL-5 locally in the bone marrow in IL-33-driven eosinophilia [22]. In this study, we determined the IL-5 production in bone marrow ILC2s and T_H_ cells in response to several stimuli for 24 h. HDM alone did not induce IL-5 production in neither bone marrow cells from control nor HDM/HDM mice. IL-5 production was not observed in T_H_ cells after IL-33 stimulation, but an increased IL-5 production was found in ILC2s (Figure 4A), which was in agreement with our previous results [22,33]. However, in comparison with IL-33 alone, the combination of HDM+IL-33 increased the number of IL-5^+^ ILC2s even further. These data suggest that there could be a potential synergistic effect of HDM and IL-33.

In a more clinical context, it is important to understand the complexity in eosinophilic diseases and increase the understanding of disease drivers and mechanisms. The IL-33/ST2 axis is indeed a potential drug target for individuals with uncontrolled eosinophilic conditions, including asthma. To conclude, bone marrow ILC2s, T_H_ cells, and mature eosinophils responded to intranasal allergen exposures of HDM by the IL-33/ST2 axis. Furthermore, our data suggest a critical role for IL-33-responsive ILC2s at the onset of allergen-induced bone marrow eosinophilia. There are currently several ongoing clinical trials targeting IL-33 or ST2 [33,49]. By targeting this axis, eosinophil levels and other inflammatory responses could potentially be reduced for individuals with still unmet clinical needs despite anti-IL-5 and other asthma therapies [31,32].

## 4. Materials and Methods

### 4.1. Mice

Male wild-type (WT) (C57BL/6J, Charles River) mice, 9–12 weeks of age were used in all experiments. The animal experiments were approved by the Gothenburg County Regional Ethical Committee (permit number 126/14; approval date 24th of June 2014). Mice were housed in pathogen-free conditions and were given food and water *ad libitum*.

### 4.2. Induction of Allergic Airway Inflammation by House Dust Mite Allergen

To induce allergic airway inflammation, WT mice were either sensitized only (HDM/PBS) or sensitized and challenged (HDM/HDM) with HDM extract (*Dermatophagoides pteronyssinus*; Stallergenes Greer, Lenoir, NC, USA) by intranasal (i.n.) administrations on days 1–3 (50 µg/day) and days 15–18 (5 µg/day), as outlined in Figure 1A. The HDM doses refer to total protein concentration of the extract and were reconstituted accordingly in 1× Dulbecco’s phosphate-buffered saline (DPBS). Control mice received the DPBS vehicle (PBS/PBS group). The allergen model was adopted from [50].

### 4.3. Sample Collection, Differential Cell Count, and Mediator Measurements

Bone marrow and BAL fluid were collected in this study as previously described [51]. BAL cells were stained (Hemacolor Rapid stain; Sigma), and cell composition was analyzed by differential cell count. Cell-free BAL was processed for mediator analysis by ELISA detecting CCL24/Eotaxin-2 (DuoSet^®^, R&D Systems, Minneapolis, MN, USA) according to the manufacturer’s instructions. Absorbance was measured on a Varioskan™ LUX multimode microplate reader (ThermoFisher Scientific, Vantaa, Finland). Bone marrow cells were obtained from femurs by flushing with wash buffer (2% fetal bovine serum, Sigma-Aldrich, 1× PBS, and filtered through a 100 µm cell strainer. Red blood cells were lysed (0.1 mM ethylenediaminetetraacetic acid in distilled water/0.8% NH_4_Cl, Sigma-Aldrich/Merck Chemicals) by 10 min incubation on ice. Then, bone marrow cells were further processed for ex vivo stimulations and flow cytometric analysis.

### 4.4. Ex Vivo Stimulations of Bone Marrow Cells

Bone marrow cells from PBS/PBS and HDM/HDM mice were seeded at a concentration of 2.5 × 10^6^/mL in complete cell culture medium: RPMI-1640 (HyClone™; GE Healthcare Life Sciences, South Logan, UT, USA), 10% fetal bovine serum (Sigma-Aldrich, St. Louis, MI, USA), 2 mM L-glutamine (HyClone), 100 U/mL penicillin, 100 µg/mL streptomycin (HyClone), and 1 mM sodium pyruvate (Sigma-Aldrich). Cells were stimulated for 24 h with either HDM (50µg/mL), rmIL-33 (PeproTech, Rocky Hill, NJ, USA, 100 ng/mL), the combination of HDM+rmIL-33, or kept in complete culture medium as a control. Monensin (BD GolgiStop™, BD Biosciences) was added to all samples (4 µl/6 mL) during the last 3 h of the incubation. Newly produced IL-5 by ILC2s (SSC^lo^Lin^−^CD45^+^CD127^+^ST2^+^) and T_H_ cells (SSC^lo^CD45^+^CD3^+^CD4^+^) was measured by intracellular flow cytometry.

### 4.5. Flow Cytometry Surface Staining

Processed bone marrow cells were resuspended in 2% mouse serum (Dako, Glostrup, Denmark). Antibodies for surface receptors were added (30 min, 4 °C) followed by washes and fixation (BD CellFix™, BD Biosciences, Erembodegem, Belgium) for 15 min in the dark at room temperature (RT). Cells were washed and analyzed on a BD FACSVerse™ Flow Cytometer with BD FACSUITE™ software (BD Biosciences). Collected data were analyzed by FlowJo Software (Tree Star Inc, Ashland, OR, USA). Linage negative cells were determined as CD45^+^CD3^−^CD45R/B220^−^CD11b^−^TER-119^−^Ly-G6/Gr1^−^CD11c^−^CD19^−^NK-1.1^−^FceR1^−^. Cell types analyzed included eosinophil progenitors (SSC^lo^CD45^+^CD34^+^IL5Rα^+^), mature eosinophils (SSC^hi^CD45^+^CD34^−^IL5Rα^lo^CCR3^+^Siglec-F^hi^), ILC2s (SSC^lo^Lin^−^CD45^+^CD127^+^CD25^+^ST2^+^), and T_H_ cells (SSC^lo^CD45^+^CD3^+^CD4^+^). All antibodies used in this study are listed in Appendix A and gating strategies are shown in the Appendix A. The IL-33 receptor (ST2) expression was estimated by MFI values. Relative MFI (rMFI) equals MFI of monoclonal antibody divided by MFI of corresponding fluorescence minus one (FMO) control.

### 4.6. Flow Cytometry Intracellular Staining

Cultured bone marrow cells were stained with surface antibodies (as described above) and fixed with 4% paraformalaldehyde (Sigma-Aldrich) in PBS for 15 min at RT (in the dark). All solutions used before fixation were supplemented with monensin (4 µl/6 mL). Cells were permeabilized with 0.1% saponin (Sigma-Aldrich) in Hank’s balanced salt solution (HyClone). Anti-IL-5 antibodies or isotype control antibodies were added, and cells were incubated for 40 min at RT (in the dark). The cells were washed and flow cytometric analysis and collected data were carried out as described in Section 4.5.

### 4.7. Statistical Analysis

Data are expressed as mean ± SEM. GraphPad Prism 8 Software (GraphPad Software Inc, La Jolla, CA, USA) was used for statistical analysis. The Kruskal–Wallis test was applied to determine the variance among three groups. The nonparametric Mann–Whitney U test was used for analysis between two independent groups if significant variance was detected by the Kruskal–Wallis test. Paired Student’s t-test was used for analysis before and after stimulations of bone marrow cultures. Statistical significance was defined as * *p* < 0.05, ** *p* < 0.01, *** *p* < 0.001 and **** *p* < 0.0001.

## Figures and Tables

**Figure 1 ijms-21-03751-f001:**
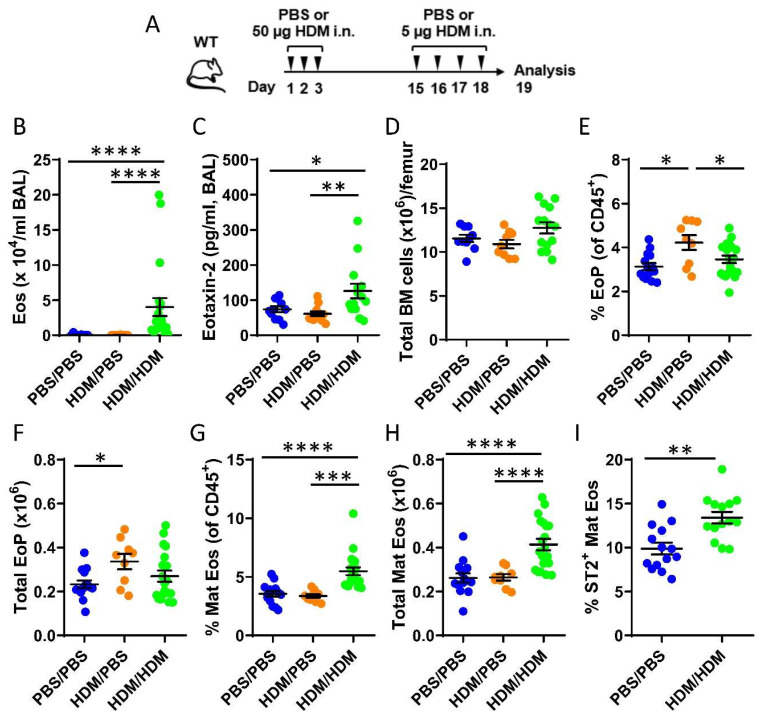
Effects of airway exposures to house dust mite (HDM) on eosinophil development and ST2 expression in the bone marrow. (**A**) Allergen inflammation model where wild type (WT) mice received intranasal (i.n.) doses of HDM extract or phosphate buffer saline (PBS) vehicle on days 1–3 and 15–18 and were sacrificed 24 h after the final exposure. (**B**) Total number of eosinophils in bronchoalveolar lavage (BAL) analyzed by differential cell count. (**C**) Concentration of CCL24/Eotaxin-2 in BAL measured by ELISA. (**D**) Total number of bone marrow cells per femur. (**E**) Number of eosinophil progenitors among all CD45^+^ bone marrow leukocytes. (**F**) Total number of eosinophil progenitors per femur. (**G**) Number of mature eosinophils among CD45^+^ bone marrow leukocytes. (**H**) Total number of mature eosinophils per femur. (**I**) Numbers of ST2^+^ Mat Eos among all CD45^+^ bone marrow leukocytes. Data are representative of 2–4 independent experiments (n = 9–20/group) and displayed as the mean ± SEM. Mann–Whitney U test. * *p* < 0.05, ** *p* < 0.01, *** *p* < 0.001 and **** *p* < 0.0001. PBS/PBS = controls. HDM/PBS = HDM-sensitized. HDM/HDM = HDM-sensitized and challenged. BM = bone marrow. EoP = eosinophil progenitors. Mat Eos = mature eosinophils. ST2 = IL-33 receptor.

**Figure 2 ijms-21-03751-f002:**
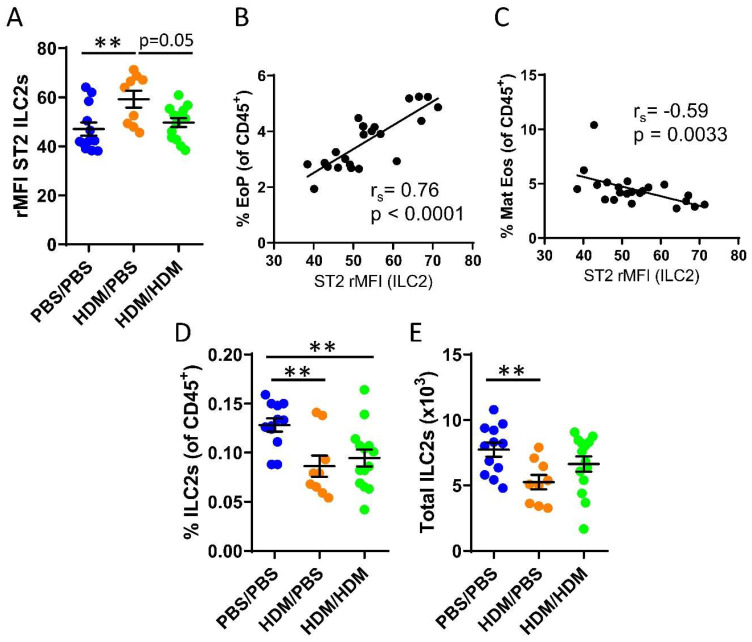
IL-33-responsive type 2 innate lymphoid cells (ILC2s) during eosinophilic inflammation in the bone marrow. (**A**) ST2 expression shown as relative mean fluorescence intensity (rMFI) on bone marrow ILC2s. Correlation plots of rMFI ST2 on ILC2s and the number of (**B**) eosinophil progenitors and (**C**) matures eosinophils in HDM/PBS and HDM/HDM mice. Number of ILC2s (**D**) among all CD45^+^ bone marrow leukocytes. (**E**) Total number of ILC2s per femur. Data are representative of 2–3 independent experiments (n = 9–14/group) and displayed as the mean ± SEM. Mann–Whitney U test. ** *p* < 0.01. Correlations using Spearman’s rho, with r_s_ indicating the Spearman correlation coefficient. PBS/PBS = controls. HDM/PBS = HDM-sensitized. HDM/HDM = HDM-sensitized and challenged. EoP = eosinophil progenitors. Mat Eos = mature eosinophils. ST2 = IL-33 receptor.

**Figure 3 ijms-21-03751-f003:**
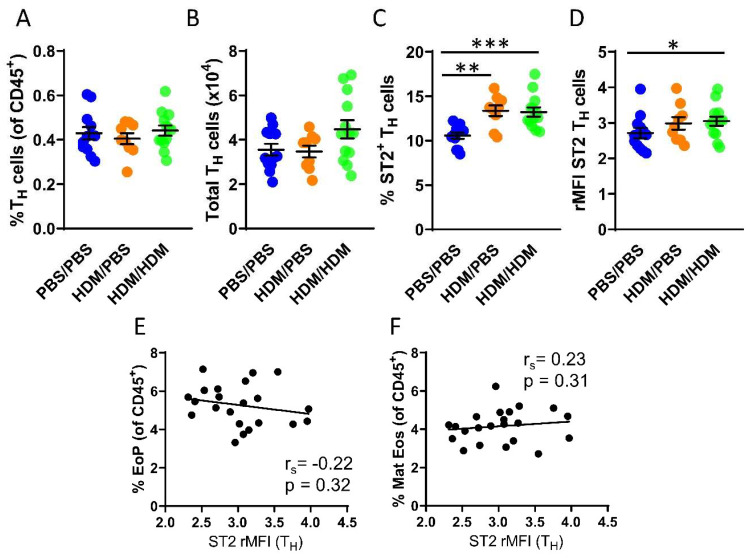
IL-33-responsive T_H_ cells during eosinophilia in the bone marrow. (**A**) Number of T_H_ cells among all CD45^+^ bone marrow leukocytes. (**B**) Total number of T_H_ cells per femur. (**C**) Number of ST2^+^ T_H_ cells. (**D**) ST2 expression on T_H_ cells shown as rMFI. Correlation plots of rMFI ST2 on T_H_ cells and the number of (**E**) eosinophil progenitors and (**F**) matures eosinophils in HDM/PBS and HDM/HDM mice. Data are representative of 2–3 independent experiments (n = 9–13/group) and displayed as the mean ± SEM. Mann-Whitney U test. * *p* < 0.05, ** *p* < 0.01 and *** *p* < 0.001. Correlations using Spearman’s rho, with r_s_ indicating the Spearman correlation coefficient. PBS/PBS = controls. HDM/PBS = HDM- sensitized. HDM/HDM = HDM-sensitized and challenged. EoP = eosinophil progenitors. Mat Eos = mature eosinophils. ST2 = IL-33 receptor.

**Figure 4 ijms-21-03751-f004:**
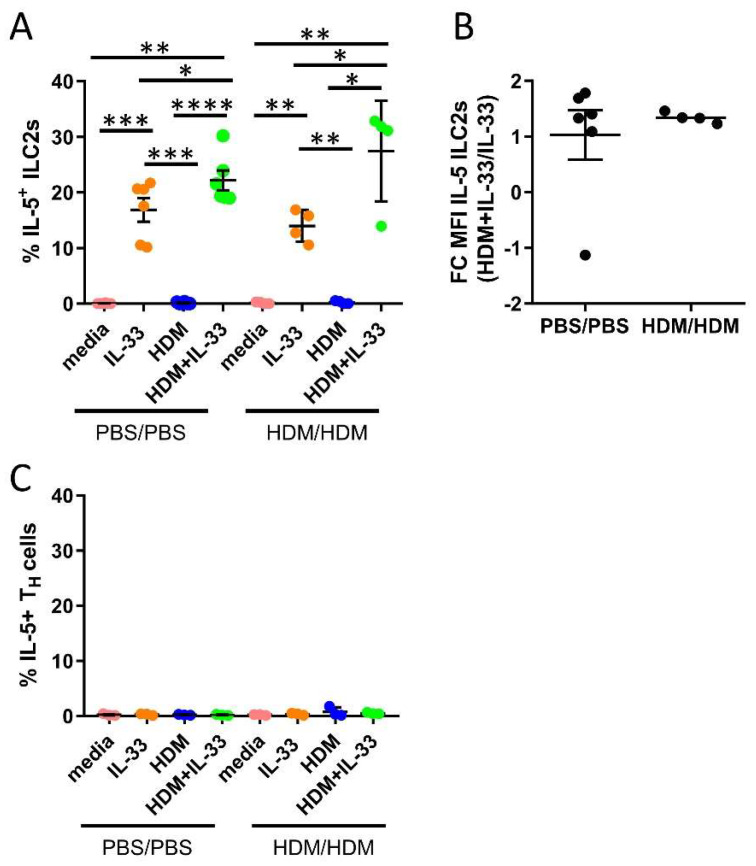
IL-5 production by bone marrow ILC2s and T_H_ cells from ex vivo stimulations for 24 h. (**A**) Number of IL-5^+^ cells among ILC2s. (**B**) Fold change (FC) mean fluorescence intensity (MFI) of IL-5 in ILC2s from HDM+IL-33 stimulated bone marrow cultures compared to IL-33 stimulated bone marrow cultures. (**C**) Number of IL-5^+^ cells among T_H_ cells. Data are representative of 1–2 independent experiments (n = 3–6/group). PBS/PBS = controls. HDM/HDM = HDM-sensitized and challenged. Media = control. Paired t-test. * *p* < 0.05, ** *p* < 0.01, *** *p* < 0.001 and **** *p* < 0.0001.

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
