# Peer review of "House Dust Mite Induces Bone Marrow IL-33-Responsive ILC2s and TH Cells"

_ijms, 2020, doi:10.3390/ijms21113751_

Round 1

Reviewer 1 Report

In this manuscript Boberg E. et al., provide evidence showing how bone marrow ILC2s, Th cells and eosinophils are regulated by IL-33 in numbers and activation state, using the aeroallergen model of house dust mite. Although there could be more proves of evidence to show the activation state and the Th cells could be also investigated in more depth, I consider this manuscript as appropriate to be published at the International journal of Molecular Sciences, as it can be very informative for researches in the field.

The only comment that I have is that the authors should include in their supplementary data figures describing their gating strategy and in the table with the antibodies to include also the dilutions used to perform the stainings.

Author Response

1: The only comment that I have is that the authors should include in their supplementary data figures describing their gating strategy and in the table with the antibodies to include also the dilutions used to perform the stainings.

Reply:

We would like to thank the reviewer for the critical review and useful comments as they have improved the manuscript. We are pleased to hear that the reviewer find the work appropriate for publication and informative for colleagues in the research field. Moreover, the gating strategy and dilution for the stainings have been added in the supplementrary material accordingly. Please find the updated version in the uploaded manuscript.

Reviewer 2 Report

This is a well written manuscript supported systematically by its data. 

What I lack to be convinced is presentation of real data rather than percentage or score. No matter what publication I looked up referring to the same topic, whether older or recent, all present real data.

Therefore please present real data where applicable.

Author Response

1: What I lack to be convinced is presentation of real data rather than percentage or score. No matter what publication I looked up referring to the same topic, whether older or recent, all present real data.

Therefore please present real data where applicable.

Reply:

We would like to thank the reviewer for the critical review and useful comments as they have improved the work. We are pleased to hear that the reviewer think it is a well written manuscript supported by the data. To address the comment regarding presentation of real data, we have now added total cell numbers where applicable in Figure 1, Figure 2 and Figure 3. Please find the updated figures with the total cell number presented in the uploaded manuscript.